Effects of Stellera chamaejasme removal on the nutrient stoichiometry of S. chamaejasme-dominated grasslands in the Qinghai–Tibetan plateau

Song Meiling 1 meilings@163.com
Wang Yuqin 1
Bao Gensheng 1
Wang Hongsheng 1
Yin Yali 1
Li Xiuzhang 1 2
Zhang Chunping 1
1 Qinghai Academy of Animal and Veterinary Sciences, State Key Laboratory of Plateau Ecology and Agriculture, Qinghai University , Xining, Qinghai , China
2 State Key Laboratory of Grassland Agro-Ecosystems, SKLGAE, Lanzhou University , Lanzhou, Gansu , China
Giordani Paolo
Electronic publication date: 2020 Jun 23
Publication date: 2020
Volume: 8
Electronic Location ID: e9239
Received 2019 Nov 14; Accepted 2020 May 5
Copyright: © 2020 Song et al.
Copyright year: 2020
Copyright holder: Song et al.
License: This is an open access article distributed under the terms of the Creative Commons Attribution License, which permits unrestricted use, distribution, reproduction and adaptation in any medium and for any purpose provided that it is properly attributed. For attribution, the original author(s), title, publication source (PeerJ) and either DOI or URL of the article must be cited.
License URL: https://creativecommons.org/licenses/by/4.0/

Keywords: Stellera chamaejasme, Plant removal, Nutrient stoichiometry, Qinghai–Tibetan Plateau, Toxic weed

Funding: Natural Science Foundation of Qinghai Province (2017-ZJ-918Q) Program for Focus on Research & Development and Transformation of Qinghai Province 2019-SF-151 National Natural Science Foundation of China 31700098, 31660690 and 31700454 State Key Laboratory of Grassland Agro-Ecosystems (Lanzhou University) This study was supported by the Natural Science Foundation of Qinghai Province (2017-ZJ-918Q), the Program for Focus on Research & Development and Transformation of Qinghai Province (2019-SF-151), the National Natural Science Foundation of China (31700098, 31660690, 31700454), and a grant from the State Key Laboratory of Grassland Agro-Ecosystems (Lanzhou University). The funders had no role in study design, data collection and analysis, decision to publish, or preparation of the manuscript.

==============================
Background

Stoichiometric relations drive powerful constraints in several fundamental ecosystem processes. However, limited studies have been conducted on the ecological stoichiometry of plants after the change of community composition induced by Stellera chamaejasme removal in alpine grassland in the Qinghai–Tibetan Plateau.

Methods

We investigated the effects of S. chamaejasme removal on ecological stoichiometry by estimating the C:N:P stoichiometry in species, functional group and community levels of the ecosystem. The interactions between different species, functional groups and correlation with soil nutrient, responding to S. chamaejasme removal were also analyzed.

Results

For the plants that became dominant after S. chamaejasme removal (SR), N content decreased and their C:N increased. S. chamaejasme removal significantly affected the nutrient stoichiometry of different functional groups. Specifically, Gramineae in the SR sites had decreased N content and N:P, and increased C:N; however, forbs had increased N content, C:P and N:P and decreased P content and C:N. At the community level, N content was lower and C:N higher in SR communities compared to CK. The N content of the plant community was positively correlated with soil total N content. S. chamaejasme removal could change the nutrient balance from species level, to functional group level, and to community level. Thus, supplementary measures might be cooperated with S. chamaejasme removal for the recovery of S. chamaejasme-dominated degraded grassland. These results provide insight into the role of S. chamaejasme in ecological protection and conservation, and the conclusions from this study could be used to develop effective and sustainable measures for S. chamaejasme control in the Qinghai–Tibetan Plateau.

Introduction

Ecological stoichiometry is used to examine the relationships between organisms and ecosystem structure and function, and reflects the dynamic balance of multiple key element, most carbon (C), nitrogen (N) and phosphorus (P), in ecological system (Elser et al., 1996, 2000; Sterner & Elser, 2002). C:N:P stoichiometry plays an important role in key ecological processes, including but not limited to, plant–herbivore–predator relationships (Kagata & Ohgushi, 2006; Tibbets & Molles, 2005), ecosystem-specific composition and diversity (Güsewell et al., 2005; Olde Venterink et al., 2003), and the capacity of a system to adapt to environmental stress (Sardans et al., 2008; Sardans, Rivas-Ubach & Peñuelas, 2012; Song et al., 2015). Many studies have shown that stoichiometric ratios at species-level are sensitive to global change drivers, such as nitrogen deposition, alteration of precipitation regime, and so on (Henry et al., 2006; Lü et al., 2012, 2018). Many plant species with various nutrient characteristics coexist in most natural ecosystems (Hou et al., 2019; Lü & Han, 2010). The responses of community-level nutrient status is simultaneously governed by variation at individual species or functional group level and alteration of community composition, as well as co-variation between them (Hu et al., 2020; Lepš et al., 2011). Stoichiometric homeostasis describes the capacity of an organism or ecosystem to maintain its internal elemental balance regardless of resource supply (Borer et al., 2015; Sterner & Elser, 2002); therefore, it is crucial to examine C:N:P stoichiometry at not only the species level but also at the community level.

Plant stoichiometric traits can be influenced by its neighboring species and the richness of the ecosystem (Abbas et al., 2013; Borer et al., 2015; Guiz et al., 2018). Because there is a wide diversity in innate characteristics between plant species, such as life-history, physiology and tissue chemistry, each has a unique influence on C, N or P cycling and their stoichiometry in an ecosystem (Ehrenfeld, 2003; Eviner, 2004; Scott, Saggar & McIntosh, 2001). Previous studies in high altitude grasslands have found that net plant–plant interactions will shift from competitive to facilitative in response to environmental change (Bret-Harte et al., 2004; Callaway et al., 2002). While plants with different life strategies will compete for limited resources (water, light and nutrients), they may also acquire facilitative shelter from their neighbors against severe climatic events such as solar radiation, strong winds and low temperature in alpine grasslands (Klanderud & Totland, 2005; Wang et al., 2008). Changes in community traits could be weighted by species relative abundance, which are more driven by dominant species rather than subdominant species in community (Violle et al., 2012; Hou et al., 2019). Thus, any changes in community composition would have implications for the changes of community level nutritional traits (Lü et al., 2018). On the other hand, many studies have reported that the species-specific interactions have influence on plant growth and community composition (Callaway et al., 2002; Wang et al., 2008). However, there has this far been little attention paid to the potential impacts of species on the nutrient cycling processes in alpine grasslands at all levels from species to functional group and community.

Aside from their innate characteristics, plants can influence C, N and P cycling and stoichiometry by modifying the biomass, composition and/or activity of the soil microbial community (Bezemer et al., 2006; Ehrenfeld, 2003; Groffman et al., 1996; Sun et al., 2009). S. chamaejasme is a toxic perennial weed found in the eastern alpine grassland of the Qinghai–Tibetan Plateau (QTP) of China. S. chamaejasme has become a dominant species, especially in heavily-grazed grassland, and can seriously threaten alpine grassland productivity and ecological sustainability (Liu, Long & Yao, 2004; Xing & Song, 2002). Thus, many research studied on S. chamaejasme exclusion and control for degraded grassland recovery (Song et al., 2018; Wang et al., 2018). S. chamaejasme spreads for many reasons including toxicity to livestock preventing its consumption (Liu, Long & Yao, 2004), its allelopathic effects on forages (Zhou, Huang & Rong, 1998), and its association with creating “fertility islands”, which enable greater soil nutrient availability (Guo & Wang, 2018; Sun et al., 2009). Hence, S. chamaejasme may induce the changes of soil nutrient or growth of other species. Although the effect of removal on plant communities has been reported, the effects on C, N and P community stoichiometry have not been studied before. The information resulting from this research might aid in understanding this species role in ecological protection and conservation in alpine grasslands.

Empirical and theoretical evidence shows that a species change would affect the community composition, and any changes in community composition would have implications for the community level nutritional traits (Callaway et al., 2002; Hou et al., 2019; Lü et al., 2018). We hypothesized that S. chamaejasme removal would induce different changes of C:N:P stoichiometry of plants from species level to functional group level, and to community level. Sun et al. (2009) reported that S. chamaejasme promoted its expansion through creating islands of fertility, which had higher N availability and turnover rates in S. chamaejasme patches soils. We further hypothesized that the concentration of N in community plants might be promoted by the higher soil N availability after S. chamaejasme removal.

Materials and Methods

Study site

The study was conducted in an alpine grassland at an elevation of 3,230 m in Haiyan County (N 37°04′, E 100°52′), approximately 125 km northwest of Xining, capital city of the Qinghai Province, China. This area has a typical plateau continental climate, with a mean annual solar radiation of 2,580 h, mean annual temperature of 0.4–3.4 °C, and annual precipitation of 277.8–499.5 mm (most of which falls between May and September). Vegetation is typical of an alpine grassland, with Kobresia and Elymus species being the dominant plants in our study area. Other companion species included Festuca ovina, Poa pratensis, Melissitus ruthenica, Kobresia humilis, Carex atrofusca and Lancea tibetica. Local herders use the study site as a winter rangeland (grazing from September to May) with a heavy grazing intensity of about 7.94 sheep units per hm2. Within the last few decades, S. chamaejasme has invaded the grassland and gradually become the dominant species in the study area resulting in the grassland facing serious degradation challenges. Field experiments were approved by the Haiyan County Grassland Station, Haibei, Qinghai (approval number: 2016-NK-136).

Experimental design

Given that the different topographical distribution, the experimental plots were arranged as a randomized blocked design with three blocks (40 × 60 m in size) located along the drainage gradient, in May 2016. Each block was 30–40 m apart. In each block, two treatments (Control, CK; S. chamaejasme removal, SR) were established with four replicate plots (20 × 7 m in size), resulting in a total of 12 plots for CK and SR respectively. In SR, S. chamaejasme were artificially removed by pulling out in June 2016, and the soil which had been carried out was returned to the original site immediately. Plots were monitored weekly during the growing season to ensure there was no further S. chamaejasme growth.

Plant sampling and chemical analysis

For species level, Elymus nutans, Poa crymophila, Koeleria litvinowii, Festuca ovina, Stipa aliena, Kobresia capillifolia, Kobresia humilis, and Carex atrofusca, which had a relative coverage of over 80%, were chosen to investigate the influence of S. chamaejasme removal on the species C:N:P stoichiometry. At the beginning of August 2017, 30 consistent leaves of each species were collected in each plot, and all leaves collected from four plots in a block were pooled as one sample, respectively. Thus, each treatment had three replications in total. For the functional group level, three quadrats (0.5 × 0.5 m) were randomly placed in each plot, and, in each quadrat, the leaves of all species were sampled and sorted into four functional groups (Gramineae, sedges, legumes and forbs). Similarly, all leaves collected from four plots in each block were pooled as one sample. For the community level, another three quadrats were randomly surveyed in each plot, and the leaves of all species were collected and pooled as one sample in each block. After sampling, all collected leaves were oven dried at 85 °C to a constant mass, and then ground for further nutrient analysis in the laboratories in Xining, Qinghai University. Soil samples were collected with a soil auger at five random sites in each plot, then the soil samples collected from four plots of each block were pooled into one sample with a separation of 0–10 cm soil depth and 10–20 cm soil depth. Soil samples were passed through a 1 mm sieve after air-drying to analyze nutrient content.

The total soil and plant organic C content was determined using the oil bath-K2CrO7 titration method—oxidization with dichromate in the presence of H2SO4, heated at 180 °C for 5 min and titration with FeSO4 (Bao, 1999). The total N content of the soil and plant samples following a Kjeldahl digestion was assayed using a Nitrogen Analyzer System (Kjeltec 2300 Auto System II, Foss Tecator AB, Höganäs, Sweden), using H2SO4 for digestion, NH3 was captured by H3BO3 and then titrated by HCl. Total P content was determined using the molybdate blue colorimetric method using a spectrophotometer (SP-723; Shanghai, China) after digestion with H2SO4 and H2O2. The levels of NH4+-N and NO3− -N in the soil samples were measured using a FIAstar 5000 Analyzer FOSS TECATOR. The available P content of the soil was analyzed according to soil agricultural chemistry methods (Bao, 1999). Stoichiometric ratios (C:N, C:P and N:P) in plant were calculated on mass basis.

Statistical analysis

Data analyses were performed using SPSS (version 17.0). For each plant sample, the C, N and P content were measured twice, thus mean values in the text are averages of six replications ± SE. Two-way ANOVA was used to determine the effects of either species and treatments, or functional groups and treatments on C, N and P levels, on the ratios of C:N, C:P and N:P, and transformed data was used, when necessary, to satisfy the assumptions of ANOVA. Independent t-tests were used to calculate significance of differences between CK and S. chamaejasme removal treatments in all parameters. Statistical significance was defined at the 95% confidence level. A principal component analysis (PCA) to assess the various effects of treatments on C, N and P levels, and the ratios of C:N, C:P and N:P in different species or functional groups were performed. A redundancy analysis (RDA) conducted in CANOCO 5.0 for Windows was utilized to assess variation ordination of community stoichiometry traits (contents of C, N and P, ratios of C:N, C:P and N:P) and soil nutrient levels (contents of Organic C, total N, total P, NH4+-N, NO3− -N, and available P in 10–20 cm deep soil).

Results

Hierarchical responses of plant stoichiometry

At species level, total C content of green leaves varied between species, but no significant difference was found between SR and CK (Fig. 1A; Table 1). In total, the treatment and species both had significant impacts on N content, but only species richness significantly altered P content (Table 1). Specifically, significantly lower N contents were observed in the green leaves of E. nutans, P. crymophila, K. litvinowii and S. aliena in SR than CK (Fig. 1C). The total P content of P. crymophila was significantly higher in SR than CK, but for C. atrofusca, the total P content was significantly lower in SR than CK (Fig. 1E). No interaction between the treatment and species was found on the C, N and P content (Table 1). However, species and treatment both significantly affected the C:N, C:P and N:P ratio in this study, except the difference was not significant for treatment on the C:P ratio (Table 1). Species and treatment interacted to affect the N:P ratio. The C:N ratio was elevated in P. crymophila, K. litvinowii, F. ovina, and S. aliena leaves in SR compared to CK, and no significant change was seen in the leaves of the other species (Fig. 1B). The C:P ratio significantly declined in P. crymophila and increased in C. atrofusca in SR compared to CK, and no significant difference was seen in the other species (Fig. 1D). The N:P ratio of P. crymophila and K. litvinowii significantly decreased in SR compared to CK (respectively), but no difference was observed in the other species (Fig. 1F).

Figure 1 Effects of Stellera chamaejasme removal on total C, N and P concentrations (A, C and E), and C:N, C:P and N:P ratios (B, D and F) of different species in an alpine grassland.

E.n: Elymus nutans; P.c: Poa crymophila; K.l: Koeleria litvinowii; F.o: Festuca ovina; S.a: Stipa aliena; K.c: Kobresia capillifolia; K.h: Kobresia humilis; C.a: Carex atrofusc. An asterisk (*) denotes a significant difference (P < 0.05).

Table 1 Results of a two-way ANOVA for the effects of species (S) and treatments (T) on the content of C, N and P and ratios of C:N, C:P, and N:P in an alpine grassland.

Items	df	C content	N content	P content	C:N ratio	C:P ratio	N:P ratio	
	F	P	F	P	F	P	F	P	F	P	F	P	
Species (S)	7	2.80	0.012	5.15	<0.001	3.61	0.002	7.50	<0.001	4.95	<0.001	3.02	0.007	
Treatments (T)	1	0.00	0.985	22.17	<0.001	0.36	0.550	22.43	<0.001	0.25	0.618	8.83	0.004	
S × T	7	0.79	0.596	0.83	0.563	2.03	0.061	1.72	0.117	2.10	0.053	3.38	0.003	

At the functional group level, total C content of green leaves varied between groups, but no significant difference was found between SR and CK. There was no significant interaction between groups and treatments in affecting C content (Fig. 2A; Table 2). Groups had significant impacts on N and P content, but only P content was significantly affected by the treatment. The interaction of groups and treatments was significant for N and P content (Table 2). Legume had the highest N content among all the functional groups, and SR treatment did not significantly affect the N content of legumes. Total N content declined significantly in Gramineae but increased significantly in forbs in SR compared to CK (Fig. 2C). Additionally, forbs had a significantly lower P content in SR compared to CK, while no significant difference was seen in Gramineae, sedges, or legumes (Fig. 2E). Groups had significant impacts on C:N and C:P ratios, but treatments had no significant impact on C:N, C:P and N:P. The interaction of groups and treatments was significant for C:N, C:P and N:P (Table 2). In Gramineae, there was no change in the C:P ratio, but the C:N significantly increased and the N:P significantly declined in SR compared to CK (Figs. 2B, 2D and 2F). In the leaves of forbs, the C:N ratio was significantly lower but the C:P and N:P ratios were significantly higher in SR compared to CK. There was no significant difference in the C:N, C:P and N:P ratios in leaves of sedges or legumes between SR and CK (Figs. 2B, 2D and 2F).

Figure 2 Effects of Stellera chamaejasme removal on total C, N and P concentrations (A, C and E), and C:N, C:P and N:P ratios (B, D and F) of different functional groups in an alpine grassland.

An asterisk (*) denotes a significant difference (P < 0.05).

Table 2 Results of a two-way ANOVA for the effects of functional groups (G) and treatments (T) on the content of C, N and P and ratios of C:N, C:P, and N:P in an alpine grassland.

Items	df	C content	N content	P content	C:N ratio	C:P ratio	N:P ratio	
F	P	F	P	F	P	F	P	F	P	F	P	
Groups (G)	3	3.35	0.046	66.79	<0.001	22.01	<0.001	43.58	<0.001	28.14	<0.001	0.68	0.577	
Treatments (T)	1	0.75	0.401	0.70	0.416	4.09	0.040	0.10	0.757	2.31	0.147	1.99	0.178	
G × T	3	0.38	0.767	7.61	0.002	4.00	0.026	5.36	0.010	3.31	0.047	5.80	0.007	

At the community level, SR significantly reduced plant total N content and increased C:N ratio (Figs. 3B and 3C). There was no significant difference in the other parameters at the community level between SR and CK (Figs. 3A and 3D–3F).

Figure 3 Effects of Stellera chamaejasme removal on total C, N and P concentrations (A, C and E), and C:N, C:P and N:P ratios (B, D and F) of the plant community in an alpine grassland.

An asterisk (*) denotes a significant difference (P < 0.05).

Driving factors of plant stoichiometry traits

The PCA analysis showed that the different species and functional groups all showed varying degrees of changes in their leaf C, N and P levels, and C:N, C:P and N:P ratios between SR and CK (Figs. 4 and 5). The first two axes of the PCA account for over 80% of the variation in species traits across the sites for all eight species, with P. crymophila and K. litvinowii showing significant differentiation in the first axis (Figs. 4A–4H). At the functional group level, all groups besides legumes showed significant differentiation, with S. chamaejasme removal responsible for over 75% variations for all four functional groups (Figs. 5A–5D).

Figure 4 Principal component analysis (PCA) of the effect of Stellera chamaejasme removal on the stoichiometric traits of different species.

(A) Elymus nutans; (B) Poa crymophila; (C) Koeleria litvinowii; (D) Festuca ovina; (E) Stipa aliena; (F) Kobresia capillifolia; (G) Kobresia humilis; (H) Carex atrofusc. A green circle (○) and green dotted line represent CK; a red triangle (Δ) and red dotted line represent SR.

Figure 5 Principal component analysis (PCA) of the effect of Stellera chamaejasme removal on the stoichiometric traits of different functional groups.

(A) Gramineae; (B) sedges; (C) legumes; (D) forbs. A green circle (○) and green dotted line represent CK; a red triangle (Δ) and red dotted line represent SR.

RDA analysis showed that approximately 80% of the variations had been explained and that S. chamaejasme removal had a significant influence on the plant N content and C:N ratio (Fig. 6). The content of organic C, total N and available P in soil were positively correlated with the N content of leaves, but negatively correlated with the C:N ratio of the community. Total P content in the soil was positively correlated with the C:P and N:P ratios of the community and negative correlation with leaf P content.

Figure 6 Redundancy analysis (RDA) of the effect of Stellera chamaejasme removal on the stoichiometric traits of the community with soil property.

A green circle (○) and green dotted line represents CK; a red triangle (Δ) and red dotted line represents SR.

Discussion

Our results showed that the responses of nutritional trait to S. chamaejasme removal were different at different biological organization levels. This was consistent with our first hypothesis. However, the N content of the community declined and the C:N ratio increased after S. chamaejasme removal, which is contrary to our second hypothesis. This may be related to the dilution effects by stimulating plant growth of some species (Sardans & Peñuelas, 2008).

In a terrestrial ecosystem, nutrient availability is one of the most limiting factors of plant growth, and thus nutrient use strategies will help determine plant distribution and dominance (Güsewell et al., 2005). The nutrient contents of green tissues could reflect the efficiency of nutrient utilization. Low nutrient concentrations in green tissues are considered to be an efficient mechanism of nutrient conservation and utilization (Carrera, Sain & Bertiller, 2000). Sistla & Schimel (2012) showed that a high C content in green tissues led to higher nutrient use efficiency. In our study, nutrient contents and C levels were species-specific (Fig. 1A). The N content of four species, E. nutans, P. crymophila, K. litvinowii and S. aliena, decreased and the C:N ratio increased following S. chamaejasme removal (Figs. 1B and 1C). This result may be explained by the increased dominance of these species (Table S1). According to this, they may have developed a N storage strategy in response to neighbor removal in which more N is transported to the reproductive organs during the reproductive growth process or to the roots before the wilt period begins. Therefore, the leaf N content was maintained at a low level in August (Rong et al., 2015). Tilman (1982) speculated that at the resource competition scale, species with low nutrient element concentrations were more suitable for growing in nutrient poor environments. In our study, the N:P ratio of most species (except P. crymophila in CK and C. atrofusca in SR) at both the CK and SR sites were lower than the threshold of 10:1 (Güsewell, 2004), suggesting that N is limited, rather than P, in this alpine grassland (Fig. 1F). This could also indicate that these species reach dominance because they have lower N needs than the others under N limited conditions. In a N poor environment, enhancing the efficiency of N utilization is an important strategy to increase species dominance, and species with lower N concentrations should have a competitive advantage over other species in N-restricted environments (Fan, Harris & Zhong, 2016; Tilman, 1997). The N:P ratio of P. crymophila in CK and C. atrofusca in SR were between 10 and 20 (Fig. 1F), which indicate that the limitation of N and P for these two species might transform because of S. chamaejasme removal. This could partly explain the increase in the P content of P. crymophila and the decrease of P content in C. atrofusca after S. chamaejasme removal, which may also indicate the strategy of these species for taking up and incorporating P element has been influenced. Overall, these results show that plants could change the nutrient utilization strategy in response to S. chamaejasme removal.

Differences in nutrient uptake and conservation strategies across growth forms and functional groups have also been previously observed (Aerts, 1996; Yuan & Chen, 2009). The nutrient element contents in plant leaves are continually affected by the plant’s structural features and growth regulation (Baldwin et al., 2006). In Gramineae, the N content decreased and the C:N ratio increased after S. chamaejasme removal (Figs. 2B and 2C). Thereby, the increase in the biomass of Gramineae may be due to their higher utilization efficiency of N and is in accordance with the “dilution theory,” where nutrient element concentration may be diluted in plant bodies when there is a rapid increase in plant biomass (Fig. S1) (Rong et al., 2015; Sardans & Peñuelas, 2008). The light:nutrient hypothesis states that the C:N ratios of plants are higher in bright environments because of the increased gains in photosynthetic C at any N concentration (Sterner et al., 1997; Sterner & Elser, 2002). Following S. chamaejasme removal in S. chamaejasme-dominated alpine grassland, environmental light levels may increase, therefore, species, such as those in Gramineae, will rapidly increase in biomass and have a lower C:N ratio. After S. chamaejasme removal, the N content increased and the C:N ratio declined in forbs, which could be explained by the utilization efficiency theory that states that a lower efficiency of N usage results in less biomass (Fig. S1). This may also be related to the increase of soil extractable inorganic N content (NH4+-N, for example, Fig. S2) after S. chamaejasme removal (Ehrenfeld, 2003), and it also indicates that the ability of forbs to absorb N, in order to maintain growth and adapt to a more severe environment, has been enhanced. The P level of an organism is partly driven by the allocation of P to ribosomal RNA, which is related to the increase in its growth rate (Hessen et al., 2007; Song et al., 2015; Vrede et al., 2004). In our study, the total P content of forbs decreased after S. chamaejasme removal (Fig. 2E), which may be partly explained by the measured decrease of the biomass of the forbs (Fig. S2). This agrees with the Growth Rate Hypothesis (GRH) that a higher plant growth rate is usually accompanied by lower C:N or C:P ratios (Elser et al., 1996; Hessen et al., 2007; Vrede et al., 2004). Previous studies in natural ecosystems have confirmed that plant biomass growth is limited by leaf N:P ratios (Das, Dang & Shivananda, 2006; Van Duren & Pegtel, 2000). The variation of autotrophs in the C:N:P composition ratio has interspecific and intraspecific components. Some analyses of the percentage of N and P of photosynthetic biomass showed that the P increased faster than N in a rapidly growing organism (Elser et al., 2000; Nielsen et al., 1996). This theory was reflected in the Gramineae, which had a faster growth rate and lower N:P ratio after S. chamaejasme removal. As for the forbs, the increased N:P ratio may be related to the increased availability of N in the soil following S. chamaejasme removal (Fig. S2). Some studies have shown that N availability increased the N:P ratio of plants (Güsewell et al., 2005), which may explain the decline of the biomass of forbs after S. chamaejasme removal (Fig. S1).

In most terrestrial ecosystems, N and P are the main elements that control plant growth (Aerts & Chapin, 2000). The stoichiometric ratios of C:N:P in plant leaves and litter in many ecosystems have been widely used as indicators to estimate nutrient limitations on plant growth, primary productivity and litter decomposition (Güsewell et al., 2005; Tessier & Raynal, 2003). Our results showed that the N:P ratio in the alpine grassland community did not change after S. chamaejasme removal (Fig. 3F), which indicates that in a short time S. chamaejasme removal has little influence on nutrient limitation in this ecosystem. The average N:P ratio at CK and SR were both <10 and therefore suggest that N was the limiting nutrient in this alpine grassland. The N content of the community declined and the C:N ratio increased after S. chamaejasme removal (Figs. 3B and 3C). This may due to the dilution effects by stimulating plant growth of several species. The aboveground biomass at SR was significantly higher than in CK (Fig. S1), indicating that the plants in this community have adapted to S. chamaejasme removal by increasing their N utilization efficiency. A previous study has shown that improving the ability to use an element in the environment where the element is limiting is important for plant growth in nutrient poor soils (Tilman, 1997). The decreases in the N utilization efficiency of forbs resulted in a decrease in biomass (Fig. S1), which helps to explain the plant community composition changes after S. chamaejasme removal. In this study, plant diversity was shown to significantly decrease after S. chamaejasme removal (Table S2). Some studies have shown that N availability increases the body N:P ratio and reduces the species diversity of communities (Güsewell et al., 2005; Roem & Berendse, 2000; Seastedt & Vaccaro, 2001). Our results showed that there was no significant change in the N:P ratio of the community between CK and SR; however, the available N (NH4+-N) content of the soil increased (Fig. S2) and the N:P ratio increased significantly after S. chamaejasme removal. The decline of community diversity seen in this study may therefore be attributed mainly to the reduction of species richness in the forbs group. From the species level to the functional group level, and then on to the community level, the variation of N and P concentrations and C:N:P ratios gradually decreased and stabilized. This may relate to grassland ecosystem stability, where the differences among species are balanced out with a greater number of species in the higher vegetation levels (Fan, Harris & Zhong, 2016). This implies that the changes of species stoichiometry induced by S. chamaejasme removal may be inhibited by inertia effects in ecosystem level.

It is well known that plants and soil are interdependent (Da Silva & Batalha, 2008) and there are an increasing number of reports that show that the nutrient traits of plants cannot be separated from the dynamics of soil nutrient availability (Eskelinen, Stark & Mannisto, 2009; Li et al., 2014). The occurrence of weeds is highly related to soil properties (Korres et al., 2017; Walter, Chritensen & Simmelsgaard, 2002). In S. chamaejasme-dominated grasslands, the organic C content in the soil decreased significantly after S. chamaejasme removal (Fig. S2). This result was consistent with previous work. For example, S. chamaejasme increases the organic C content of soil because of the greater plant production and litter input or the higher microbial biomass (Sun et al., 2009). The results of this study showed that the organic C content in soil without S. chamaejasme was lower than that with a S. chamaejasme community. After S. chamaejasme removal, the total N, P and available P content of the soil decreased (Figs. 5C, 5E and 5F). This may be due to the fast growth rate of the grasses, which requires greater N and P uptake in the absence of competition from S. chamaejasme. The content of NH4+-N in soil increased significantly after S. chamaejasme removal (Fig. S2). S. chamaejasme is widely distributed throughout alpine grasslands creating islands of fertility, as determined by greater soil nutrient availability (Sun et al., 2009). Therefore, when S. chamaejasme has been removed, the soil nutrients gathered by the plants may be released from these “fertility islands,” (Guo & Wang, 2018; Sun et al., 2009), which was reflected by the change of NH4+-N content observed in the results. The leaf trait–environment relationship is used to explain and predict the underlying mechanisms of leaf nutrient trait variation, environmental change, and to identify the nutrient limitations in an ecosystem (Güsewell, 2004; Kerkhoff et al., 2005; Zhang et al., 2019).

Conclusions

Different species and functional groups were shown to have different responses to S. chamaejasme removal as seen in their N and P levels and C:N, C:P and N:P ratios. The species with higher aboveground biomass or dominance has lower N content (like Gramineae), but the species with lower biomass has higher N content and lower P content (like forbs), after S. chamaejasme removal. On the community level, the N content was lower and the C:N ratio was higher in S. chamaejasme removal site than in control site. Thus, S. chamaejasme removal can cause imbalances in the C:N:P stoichiometric ratio as shown in this ecosystem during the period of experiment, and then change species dominance, composition or diversity of the community. Other measures, like supplementary sowing of dicotyledon species, appropriate grazing at growing season, or fertilization, cooperated with S. chamaejasme removal may be better for conservation and preservation of the function and species richness in S. chamaejasme-dominated grassland. This study presents the analysis of results obtained from one year of data collection. More systematic studies need to be carried out in alpine S. chamaejasme-dominated grasslands in order to reveal the influence of longer time periods and other factors, such as microorganisms, climate, and grazing, or the impact of other alternative measures such as fertilization and grazing prohibition on the recovery of S. chamaejasme-dominated degraded grassland.

Supplemental Information

Supplemental Information 1 Effects of Stellera chamaejasme removal on aboveground biomass of (a) different species and (b) functional groups in an alpine grassland.

E.n: Elymus nutans; P.c: Poa crymophila; K.l: Koeleria litvinowii; F.o: Festuca ovina; S.a: Stipa aliena; K.c: Kobresia capillifolia; K.h: Kobresia humilis; C.a: Carex atrofusc.Asterisk (*) denotes a significant difference (P < 0.05).

Click here for additional data file.

Supplemental Information 2 Effects of Stellera chamaejasme removal on organic C (A), NH4+-N (B), total N (C), NO3--N (D), total P (E), and available P (F) concentrations in 0–10 cm and 10–20 cm deep soils in an alpine grassland.

Asterisk (*) denotes a significant difference (P < 0.05).

Click here for additional data file.

Supplemental Information 3 Raw data.

Click here for additional data file.

Supplemental Information 4 Differences in the importance value between the control (CK) and Stellera chamaejasme removal (SR) treatments.

Value shows mean ± SE. Different lowercase letters denote significant differences (P < 0.05) between CK and SR treatments.

Click here for additional data file.

Supplemental Information 5 Difference in species richness, Shannon-Wiener index, Simpson index, and Pielou index between control (CK) and S. chamaejasme removal (SR) treatments.

Different lowercase letters denote significant differences (P < 0.05) between CK and SR treatments.

Click here for additional data file.

The authors would like to thank Zengzeng Yang, Zengtao Qi, and Yan Liu for their help collecting samples and analyzing data. We also thank the editor from MogoEdit for the language modification. At last, we appreciate Dr. Alison Haughton, editor in chief from Weed Research, for the constructive advice on the manuscript during the revising process.

Additional Information and Declarations

Competing Interests

Author Contributions

Field Study Permissions

Data Availability

The authors declare that they have no competing interests.

Meiling Song conceived and designed the experiments, performed the experiments, analyzed the data, prepared figures and/or tables, and approved the final draft.

Yuqin Wang performed the experiments, prepared figures and/or tables, and approved the final draft.

Gensheng Bao conceived and designed the experiments, prepared figures and/or tables, and approved the final draft.

Hongsheng Wang performed the experiments, authored or reviewed drafts of the paper, and approved the final draft.

Yali Yin analyzed the data, authored or reviewed drafts of the paper, and approved the final draft.

Xiuzhang Li analyzed the data, prepared figures and/or tables, and approved the final draft.

Chunping Zhang analyzed the data, authored or reviewed drafts of the paper, and approved the final draft.

The following information was supplied relating to field study approvals (i.e., approving body and any reference numbers):

Field experiments were approved by the Haiyan County Grassland Station, Haibei, Qinghai (2016-NK-136).

The following information was supplied regarding data availability:

The raw measurements are available in Data S1.

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
