# Peer review of "Effects of Stellera chamaejasme removal on the nutrient stoichiometry of S. chamaejasme-dominated grasslands in the Qinghai–Tibetan plateau"

_PeerJ, doi:10.7717/peerj.9239_

## Round 0.1 · original submission · Major Revisions

Both reviewers find that your work is very interesting and that it brings important news in the field of ecological stoichiometry.

However, the reviewers make several constructive suggestions that will need to be considered in preparing the new version of the manuscript. In particular, they make some observations and concerns regarding the formulated hypotheses and research questions, the experimental design and the overall conclusions.

Details are shown in the reports below.

·

Basic reporting

This artical met the standards of PeerJ. This work would be improved by concepting a core idea. Authors would be benefitted to read several related papers which reported the stoichiometric responses of different levles ranging from species to community.
For instance, Hu et al. 2019. Legacy effects of nitrogen deposition on plant stoichiometry in a temperate grassland. Plant and Soil.
Hou et al. 2019. The relative contributions of intra- and inter-specific variation in driving community stoichiometric responses to nitrogen deposition and mowing in a grassland. Science of the Total Environment, 666: 887-893.
Wang et al. 2018. The impacts of nitrogen deposition on community N:P stoichiometry do not depend on phosphorus availability in a temperate meadow steppe. Environmental Pollution, 242: 82-89.
Lu et al. 2018. Intraspecific variation drives community-level stoichiometric responses to nitrogen and water enrichment in a temperate steppe. Plant and Soil. 423: 307-315.
Hou et al. 2018. Consistent responses of litter stoichiometry to N addition across different biological organization levels in a semi-arid grassland. Plant and Soil, 421:191-202.
Han et al. 2014. Hierarchical responses of plant stoichiometry to nitrogen deposition and mowing in a temperate steppe. Plant and Soil, 382: 175-187.

Experimental design

Experimental design is well.

Validity of the findings

Findings are valid and important.

Additional comments

The paper from Song et al. reported the ecological stoichiometric responses of plants from species level to community level to a weed removal in an alpine grassland. Findings from this work would improve our understanding of the grassland resotration processes. While the experiment is well carried out and data were well analyzed, the mnuscript would be improved by further revision.

First, the hypotheses should be improved because they are not clear at the present version. Further, the article should be written with hypotheses as the main core. The Introduction section should be provide direct background information for each hypothesis.

Second, the Introduction section should be rewritten as there is too much too general information about what is stoichiometry and the importance stoichiometry. In contrast, the introduction should be more specific and focus on the particular question you want to answer in this study.

Third, the discussion is too long. You don't have to mention every result found in this study. Instead, you should focus on the most important one and those are directly related with your hypotheses.

I hope the literature I suggested could help authors to learn how to organize the writing of your paper.

Reviewer 2 ·

Basic reporting

I would suggest choosing only a handful of all figures presented, as they are many. Pick the most important ones, and place the others into the supplementary files section. Also consider unifying the tables, in cases where it makes sense, and also move some of them into the Supplementary section.

Experimental design

Lines 118-121: I find the formulation of the hypotheses a little bit vague, and very general. I would suggest formulating the hypotheses a little bit more specific. For instance in the case of the first hypothesis: how do you expect the stoichiometric traits in foliar biomass to change, exactly?

I have general concerns about the presentation of the experimental design, as from the written description it has not become completely clear to me, how the sites were located relative to each other. First, it does not become clear whether the three study sites of 60x40m were located on three different pastures, or on the same one. Second, it did not become clear to me why the four subplots per plot-half were chosen (were the entire half plots treated by Stellera removal, or only the four suplots? And where exactly were the three sub-sub-plots for biomass and community diversity assessment located? Three per each subplot, meaning 12 per half-plot? Or three per half-plot? For this, I would suggest providing both a figure of the general location of the study sites in the Tibetan Plateau or China (optional, but would be nice) on one hand, and a Figure depicting the experimental study design for clarification (definitely important if someone wants to replicate this experiment).
Furthermore, I do not understand how the surveyed plots were exactly analyzed, or pooled. If there were three plots per half-plot, in the RDA or PCA analysis there should be 9 points per treatment visible. If there were three sub-sub-plots per subplot per half-plot, there should be 36. However, there are only 6 points visible. How was the data assessed here, pooled?

Validity of the findings

The conclusions section is basically just a repetition of all results; but the exact meaning or the utility of the results is not made clear. The use of this study in the terms of conservation and preservation of grasslands and species richness, as introduced in the abstract, is lacking though. Maybe use the conclusions for the elaboration of the possible application of your results rather than repeating results.

Additional comments

Dear authors,

I found this manuscript very interesting and very well written. I congratulate the authors on their excellent introduction to this complicated subject, rendering this subject easily understandable even for people unfamiliar with Ecological Stoichiometry.
Overall, I believe this study makes a good contribution on basic research of grassland ecology, both on a local or regional scale (alpine grasslands in the Tibetan Plateau) as well as on global scale. It makes a good contribution to studies assessing the underlying mechanisms driving species occurrence and performance, as well as community dynamics, considering the impact of dominant toxic weeds invading heavily grazed grasslands.
All underlying data has been provided and seems sound and well funded. This research was very well referenced to literature; results and conclusions were sound and well supporting each other. Figures and tables are relevant and supporting the presented study. Conclusions were well stated, linked to the original research questions and limited to the supporting results.
However, I have some suggestions and concerns regarding the formulated hypotheses and research questions, the experimental design and the overall conclusions posed by the authors, as stated below. I believe that this manuscript well fits the scope of the PeerJ Journal, and will be fit for publication once the issues presented below are addressed.

General comments:
Abstract:
Line 14 and 28: “Controlling Stellera chamaejasme is one of the main methods used to recover degraded grasslands in the Qinghai-Tibetan Plateau. …. In addition, examination of plant performance showed an increase in biomass and decline in community diversity after S. chamaejasme removal.” In the abstract, it does not immediately become clear why, considering biodiversity conservation, one would want to control a species in grazed grasslands that maintains or increases community diversity if present. When reading the introduction (Line 94), it becomes clear that these grasslands are used by herders to maintain sheep, and that the recovery of these more species rich grasslands by removing Stellera has predominatly economic reasons. I think it is necessary to elaborate why this recovery, i.e. removal of this toxic weed, is of interest, even if it causes species diversity to decrease.
Introduction:
Lines 118-121: I find the formulation of the hypotheses a little bit vague, and very general. I would suggest formulating the hypotheses a little bit more specific. For instance in the case of the first hypothesis: how do you expect the stoichiometric traits in foliar biomass to change, exactly?
Material and Methods:
I have general concerns about the presentation of the experimental design, as from the written description it has not become completely clear to me, how the sites were located relative to each other. First, it does not become clear whether the three study sites of 60x40m were located on three different pastures, or on the same one. Second, it did not become clear to me why the four subplots per plot-half were chosen (were the entire half plots treated by Stellera removal, or only the four suplots? And where exactly were the three sub-sub-plots for biomass and community diversity assessment located? Three per each subplot, meaning 12 per half-plot? Or three per half-plot? For this, I would suggest providing both a figure of the general location of the study sites in the Tibetan Plateau or China (optional, but would be nice) on one hand, and a Figure depicting the experimental study design for clarification (definitely important if someone wants to replicate this experiment).
Furthermore, I do not understand how the surveyed plots were exactly analyzed, or pooled. If there were three plots per half-plot, in the RDA or PCA analysis there should be 9 points per treatment visible. If there were three sub-sub-plots per subplot per half-plot, there should be 36. However, there are only 6 points visible. How was the data assessed here, pooled?
Just out of interest, I was wondering about the stoichiometric regulation of plant species. The authors elaborate about species-specific nutrient uptake and use efficiency by comparing mean values of nutrient content and nutrient ratios. Why did they not calculate the 1/H homeostatic regulation coefficient, which is the slope of the log (nutrient in plant biomass) ~ log (nutrient in soil) relationship, and which directly allows the estimation of excess nutrient uptake, degree of C assimilation relative to nutrient content and degree of nutrient homeostasis is species? This might be a nice addition to this study.
Also, I would suggest choosing only a handful of all figures presented, as they are many. Pick the most important ones, and place the others into the supplementary files section. Also consider unifying the tables, in cases where it makes sense, and also move some of them into the Supplementary section.

Conclusions:
The conclusions section is basically just a repetition of all results; but the exact meaning or the utility of the results is not made clear. The use of this study in the terms of conservation and preservation of grasslands and species richness, as introduced in the abstract, is lacking though. Maybe use the conclusions for the elaboration of the possible application of your results rather than repeating results.

Introduction:
Lines 61, 103: Authors mention “scholars” having examined different aspects of the surveyed grasslands. I find this terminology unsuitable, as I suppose these studies are research studies well supporting the presented one. Better refer to them as such.
Line 83: ….(water, light and nutrients)….
Line 87: …, there has this far been little attention….
Line 92: …. , and CO2, they are fierce competitors (Baker…..)” … Competitors to what? Each other? Other plants?
Line 107: “….. Most research on S. chamaejasme has focus on population spread dynamics….” I find this formulation unusual. Population expansion maybe?
Line 109: “… and the use of its extracted flavonoids”…. – what does this mean, I do not understand this? Does this mean that flavonoids have been extracted from this species´ plant biomass and spread on other plants, in order to assess their response?
Line 109: …Little is therefore known about the effects on S. chamaejasme removal on ecological stoichiometry. “ – I would consider changing the wording here to clarify this sentence. Maybe like this: Hence, although the effects of removal on plant communities has been studied before, the effects of CNP community stoichiometry have not been studied before. The information resulting from this research might aid in understandig this species role in ecological protection...

Material and Methods:
Line 129: “….Vegetation is typical of an alpine grassland, with Kobresia and Elymus species…” - add a comma
Line 131: “…Kobresia humilis, Carex atrofusca, and Lancea tibetica. ….” – remove comma
Lines 135-136: “… Field experiments were approved by …” – I think this is not a good place to state the approval of the Haiyan County grassland Station. Usually, one states the grants and fundings of a study at the end of the manuscript, at the end of the Acknowledgement-Section.

Line 147…”Heights of all living plants…” - I believe you refer to average vegetation height? Better change wording
Line 150: …”Biomass was harvested by hand clipping to ground level, sorted by species, oven dried….” . Do you refer to all species in the entire 0.5 x 0.5 m square, or only to those dominant ones that were later on analyzed?
158: …” 30 consistent leaves of each species were collected and mixed as one sample…” – I believe the correct wording here is “ were collected and pooled”. You sampled the 30 leaves from different individuals, I suppose? Clarify.

Line 191: …” and N:P in different species or functional groups was performed.” Incomplete sentence

Results
Line 200: “ Plant aboveground biomass in SR…” You refer to the total aboveground biomass of the sampled species? Or total aboveground biomass of the entire community?
Line 204: “…. On a species level, the aboveground biomass….”
Line 207: “…. K. humilis, and C. atrofusca was not different between CK…”
Line 209: “…. Shannon-Wiender index and Simpson index, respectively.”
Line 213: “… On a species level, total C content….”
Line 216: ”… but only species richness significantly altered….”
Line 223: “.. Hovewer, species diversity and treatment both….”
Line 279: “…. Across the sites for all eight species, with P. crymophila…”
Line 298: “… Besides K. humilis and C. atrofusca, all importance values of the other studied species increased…..” - what are these “importance values”? What do they mean and how are they calculated or assessed. I believe this was not explained or mentioned in the Methods section.
Line 301: “... S. chamaejasme removal may significantly enhance the competitive potential of these species via increasing available, space, which allows them to inhibit the growth of their competitors.” This may need a bit more explanation, maybe like such: “Stellera removal causes open gaps in the vegetation, which is quickly occupied by Species XX; leading to competitive exclusion of the other co-occurring species.”.
Line 314: “…. This result may help explain the increased dominance of these six species and also suggests that they may have developed a N storage strategy in response to neighbor removal, in which more N is transported to….” – This is speculative. The results do not suggest this, but this may be a possibility to explain the observed patterns. It furthermore could possibly be that species to not only store nutrients, but that they also take up more nutrients than needed (excess uptake). This entire section must be formulated in a way that clearly identifies this as speculative. “Suggest” is, form my point of view, more that speculative.
Line 323: …” N is limited, rather than P, in this alpine grassland. “ This could also indicate that these species reach dominance because they have lower N needs than the others under N limited conditions.
Line 328: … This could partly explain the increase in the P content of P. crymophila and the decrease of P content in….” - This aspect is unclear to me. Nutrient limitation is apparently species specific.
If one species takes up phosphorus under Stellera removal and increases nutrient content in its biomass - and the other species does not… how is this indicative of nutrient co-limitation? In the case of P. crymophila this seems reasonable - P. crymophila takes up P and incorporates it directly via C assimilation, maintaining high CP values. But C. atrofusca does not. Wouldn’t a species take up P and N simultaneously and proportionally under NP co-limitation?
Line 340: “…. The growth of forbs may be inhibited by Graminae plants because of their greater competitive ability.” – How could this protective mechanism work? Why could graminoids be less sensitive to environmental factors than forbs? Maybe explain this a little.
Line 405: “… From the population level to the functional group level…” – you mean species level?
Line 407: “… may relate to grassland ecosystem homeostasis, …“ – I am unsure about using this terminology here. A species can be homeostatic about its internal nutrient content, but can a whole community?
Line 410: “… Stoichiometry may be inhibited by inertia effects. “ – This is not explained properly, what does this mean?
Lie 413: “…increasing number of reports that show that the nutrient traits of plants ….”

---

## Round 0.2 · Minor Revisions

The reviewer and I agree that the quality of this paper has been improved after the revision. Some minor changes are still required.

·

Basic reporting

The quality of this work has substantially improved after revision. I suggest this work could be accepted after minor revision.

Experimental design

The experiment was well designed and carried out.

Validity of the findings

The novelty of results in this study is high.

Additional comments

Findings from this study help improve our understanding of stoichiometric changes of grassland ecosystem to invasive species removal. The quality of this paper has been improve during the revision. I suggest authors do minor revision before this paper could be accepted for publication.

1. Lines 13-16 It is important to show why studies on ecological stoichiometry are important.
2. Line 17 'We' instead of 'This study'
3. Line 21 delete 'Our results showed that'
4. Line 25-26 delete 'when compared to those at the control site (CK)'
5. Line 29 delete 'Overall, this study found that'
6. Line 33 'could' instead of 'will'
7. Line 62 'in response to'
8. Line70 'species-specific'
9. Line 167 'At species level'
10. Line 174 did you use 'species diversity' as an variable in your experiment?
11. Line 176 'to affect' instead of ', affecting'
12. Line 183 'At the functional group level'
13. Line 185 'in affecting C content'
14. Line 187-188 'Legume had the highest N content among all the functional groups.'
15. Line 199-200 'At the community level, SR significantly reduced plant total N content and increased C:N ratio.'
16. Line 217-219 this sentence could be deleted without any loss of information.
17. Line 220 'at different biological organization levels'

---

## Round 0.3 · accepted · Accept

Thank you for modifying the minor changes requested by the reviewer. It seems to me that your manuscript is now suitable for publication.